# A Novel Smart Assistance System for Blood Vessel Approaching: A Technical Report Based on Oximetry

**DOI:** 10.3390/s20071891

**Published:** 2020-03-29

**Authors:** Chien-Ching Lee, Chia-Chun Chuang, Bo-Cheng Lai, Yi-Chia Huang, Jen-Yin Chen, Bor-Shyh Lin

**Affiliations:** 1Institute of Imaging and Biomedical Photonics, National Chiao Tung University, Tainan 71150, Taiwan; otzison.cop03g@nctu.edu.tw (C.-C.L.); jack200242149.cop03g@nctu.edu.tw (B.-C.L.); reda36204.cop07g@nctu.edu.tw (Y.-C.H.); 2Department of Anesthesia, An Nan Hospital, China Medical University, Tainan 70965, Taiwan; d73088@mail.tmanh.org.tw; 3Department of Medical Sciences Industry, Chang Jung Christian University, Tainan 71101, Taiwan; 4Department of Anesthesiology, Chimei Medical Center, Tainan 71004, Taiwan; cmh7760@mail.chimei.org.tw

**Keywords:** catheter placement, anesthesia, absorption spectra, tissue components, optical density change, hemoglobin concentration

## Abstract

In clinical practice, the catheter has to be placed at an accurate position during anesthesia administration. However, effectively guiding the catheter to the accurate position in deeper tissues can be difficult for an inexperienced practitioner. We aimed to address the current issues associated with catheter placement using a novel smart assistance system for blood vessel catheter placement. We used a hollow introducer needle embedded with dual wavelength (690 and 850 nm) optical fibers to advance the tip into the subclavian vessels in anesthetized piglets. The results showed average optical density changes, and the difference between the absorption spectra and hemoglobin concentrations of different tissue components effectively identified different tissues (p < 0.05). The radial basis function neural network (RBFNN) technique was applied to distinguish tissue components (the F-measure value and accuracy were 93.02% and 94%, respectively). Finally, animal experiments were designed to validate the performance of the proposed system. Using this system based on oximetry, we easily navigated the needle tip to the target vessel. Based on the experimental results, the proposed system could effectively distinguish different tissue layers of the animals.

## 1. Introduction

Catheter placement, such as in internal jugular vein cannulation, cricothyroidotomy, and subclavian vein (SCV) cannulation, requires accurate needle placement into the target location. Anatomical landmarks and palpation are traditional methods used to guide the needle into the target location [1]. In clinical application, this is an invasive procedure, whereby we first identify the central blood vessel (most commonly the internal jugular, subclavian, or femoral vein) and then insert a catheter into it. However, these methods tend to be difficult for an inexperienced practitioner and may be influenced by the patient’s anatomy [2]. Lack of experience may cause tissue puncture during the procedure of cannulation. Therefore, to avoid this risk, a reliable method of locating the needle tip position is needed.

Although the ultrasound technique to guide cannulation has improved the success rate and decreased complications [3,4,5], it still has some limitations. When the needle is in front of or behind the narrow cross-sectional plane, in the deep three-dimensional space, the tip location of the introducer needle may be hard to identify, and the image resolution and quality will also become poorer in conjunction with the penetrating depth. Moreover, when the anatomical structure is close to the bones, the image may be influenced by the artifacts of bone shadows. Thus, to address these issues, echogenic and optic fiber techniques have been proposed to improve the ultrasonic visibility of medical devices in recent years. Xia et al. integrated a miniature optic fiber ultrasound hydrophone into a needle tip and used an ultrasound receiver in the skin surface to visualize the needle tip [6,7]. An echogenic needle [8] has also been used to improve needle tip visibility so that its surface is modified to increase backscattering toward the ultrasound imaging probe. However, for a larger angle (> 50°), the severe image artifacts still disturb the procedure. Klein et al. combined two 1–8 kHz piezoelectric actuators with a needle tip to provide the location message of the needle tip when performing ultrasound-guided regional anesthesia. However, another study evaluated that its usefulness and safety in live tissues need to be investigated [9].

The recently developed vascular transillumination device uses near-infrared light, which is absorbed by blood and reflected by surrounding tissue, to generate a two-dimensional vascular image. This technology was also developed and used to assist in superficial vein visualization [10,11,12]. The received information is processed and projected directly onto the skin surface to provide an accurate location image of superficial vessels. It is useful in assisting the operator in placing the needle accurately and reducing the length of time required for superficial vascular access [13], but it cannot monitor the deeper vessels.

In this study, we proposed a novel smart assistance system based on the differences among the absorbing spectra and the blood vessel distribution densities of various tissue components [14,15], with the aim of improving blood vessel catheter placement. Neural network technology was also applied to identify various tissue layers.

## 2. Materials and Methods

### 2.1. Design of the Smart Assistance System for Blood Vessel Catheter Placement

The smart assistance system for blood vessel catheter placement was designed to distinguish the human tissue type near the location of the needle tip, and its basic scheme and photograph are displayed in Figure 1a,b, respectively. It mainly contains three components: an optical probe, a wireless signal processing module, and a host system.

The optical probe consists of two laser diodes with wavelengths of 690 nm or 850 nm (HL6738MG, HITACHI, Japan; L850P030, THORLABS, USA), a photodiode (PD15-22C, EVERLIGHT, Taiwan), optical fibers, and a needle tube, and it is designed to provide a multi-wavelength light source and also receive the light penetrating through the tissue. Here, these optical fibers are used to transmit light into the human tissue from laser diodes, and the other optical fiber is used to receive the light penetrating through the tissue. These optical fibers are embedded into the needle tube, and they can reach the deeper tissue easily via insertion of the needle tube.

The block diagram of the wireless signal processing module is shown in Figure 2, and it contains a laser diode driver circuit, a photodiode amplifier circuit, a wireless transmission circuit, and a microprocessor. In the laser diode driver circuit, a boost converter is used to provide a high forward voltage to the laser diode. The design of the photodiode amplifier circuit is based on a trans-impedance amplifier; it can convert the photocurrent generated from the photodiode to a voltage and amplify this voltage. Then, the received voltage signal of the photodiode is digitized by a 12-bit analogue-to-digital converter built in the microprocessor (MSP430, Texas Instruments, USA) with a sample rate of 100 Hz, and is then sent to the wireless transmission circuit to transmit to the host system wirelessly via Bluetooth. The photodiode is the detector of the optical probe, which receives a few scattered or reflected photons from two laser diodes. The design of the wireless transmission circuit is compatible with the Bluetooth v2.0 specification. This module can be operated by a 3.7 V 1200 mA⁄h Li-Ion battery. The response time of the proposed system for recognizing the tissue at the needle tip was less than 20 milliseconds.

The host system was designed on a commercial laptop with a Windows 10 operating system, and in this host system, a real-time monitoring program was developed using Microsoft Visual C#. The real-time monitoring program receives the optical signal acquired from the wireless signal processing module and then calculates the relative hemoglobin concentration and tissue oxygen saturation ($StO_{2}$) from the change in optical density using the modified Beer–Lambert law (MBLL). Finally, the received optical signal and the estimated hemoglobin information are displayed and stored. The technique was also implemented for neural networks in the real-time monitoring program. In this study, radial basis function neural network (RBFNN) [16] was used to recognize the human tissue type. RBFNN is a good approximation approach for complex models. Compared with other neural networks, it has several advantages, such as a simpler network configuration, faster training procedure, and good approximation capability, and it has been widely used in many medical applications [17,18,19,20,21]. The basic framework of RBFNN includes three layers (a hidden layer, an input layer, and an output layer). In this study, a self-organized learning procedure (k-mean clustering algorithm) was used to train the center vectors of neurons in the hidden layer, and a normalized least mean square algorithm was used to train the weight vector between the hidden neurons and the output neuron [22,23]. The center vectors of the hidden neurons were read as the feature vectors extracted from the training sets.

### 2.2. Fundamental Principle of the Modified Beer–Lambert Law

The design of the proposed system is based on the modified Beer–Lambert law, which describes the light attenuation in a high scattering medium, such as human tissue [24]. When photons with a wavelength of 690 nm or 850 nm penetrate the tissue layer, some of them will be absorbed or scattered to cause light attenuation. The change in optical density (∆OD) [25] can be expressed as:(1)$$\Delta \mathrm{OD}\left(\lambda \right)=-\log\frac{I_{o}\left(\lambda \right)}{I_{i}\left(\lambda \right)}=\varepsilon CLB\left(\lambda \right)$$
where $I_{i}\left(\lambda \right)$ and $I_{o}\left(\lambda \right)$ denote the incident and penetrating light corresponding to the wavelength λ, respectively. The parameters ε, C, and L are the molar extinction coefficient, molar concentration, and the distance between the source and detector, respectively. The parameter B(λ) [26] is the differential path length factor (DPF) corresponding to the wavelength λ, which is used to correct the path length from the source to detector. Hemoglobin is one of the major absorbers of red and near-infrared light in human tissue. Therefore, for red and near-infrared light, the change in optical density can be simplified as:(2)$$\Delta \mathrm{OD}\left(\lambda \right)=\left[\varepsilon_{HbO_{2}}\left(\lambda \right)\times \left[HbO_{2}\right]+\varepsilon_{Hb}\left(\lambda \right)\times \left[Hb\right]\right]LB\left(\lambda \right)$$

Here, $\left[HbO_{2}\right]$ and [Hb] denote the relative oxy-hemoglobin ($HbO_{2}$) and deoxy-hemoglobin (Hb) concentrations, respectively, and $\varepsilon_{HbO_{2}}\left(\lambda \right)$ and $\varepsilon_{Hb}\left(\lambda \right)$ are the molar attenuation coefficients of $HbO_{2}$ and Hb corresponding to the wavelength *λ*.

According to the difference between the absorption spectra of $HbO_{2}$ and Hb, the relative $HbO_{2}$ and Hb concentrations can then be estimated from the optical density change corresponding to two or more wavelengths. For dual wavelengths of $\lambda_{1}$ and $\lambda_{2}$, the estimated $HbO_{2}$ and Hb concentrations are calculated as follows:(3)$$\left[HbO_{2}\right]=\left(\varepsilon_{Hb}\left(\lambda_{2}\right)\times \frac{\Delta \mathrm{OD}\left(\lambda_{1}\right)}{B\left(\lambda_{1}\right)}-\varepsilon_{Hb}\left(\lambda_{1}\right)\times \frac{\Delta \mathrm{OD}\left(\lambda_{2}\right)}{B\left(\lambda_{2}\right)}\right)\times \frac{1}{\det\left(A\right)}\times \frac{1}{L}$$
(4)$$\left[Hb\right]=\left(\varepsilon_{HbO_{2}}\left(\lambda_{1}\right)\times \frac{\Delta \mathrm{OD}\left(\lambda_{2}\right)}{B\left(\lambda_{2}\right)}-\varepsilon_{HbO_{2}}\left(\lambda_{2}\right)\times \frac{\Delta \mathrm{OD}\left(\lambda_{1}\right)}{B\left(\lambda_{1}\right)}\right)\times \frac{1}{\det\left(A\right)}\times \frac{1}{L}$$

Here, det(A) is the determinant of the matrix $A=\left[\begin{array}{cc} \varepsilon_{HbO_{2}}\left(\lambda_{1}\right) & \varepsilon_{Hb}\left(\lambda_{1}\right)\\ \varepsilon_{HbO_{2}}\left(\lambda_{2}\right) & \varepsilon_{Hb}\left(\lambda_{2}\right) \end{array}\right]$.

For the absorbing spectra of $HbO_{2}$ and Hb, the isosbestic point of their absorbing spectra is at about 800 nm. Therefore, 690 nm light and 850 nm light were used as the light source in this study. After estimating the relative $HbO_{2}$ and Hb concentrations, the total hemoglobin concentration [HbT] and the tissue oxygen saturation $StO_{2}$ can then be calculated as follows:(5)$$\left[HbT\right]=\left[HbO_{2}\right]+\left[Hb\right]$$
(6)$$StO_{2}=\frac{\left[HbO_{2}\right]}{\left[HbT\right]}\times 100\%$$

### 2.3. Experimental Design

The animal use protocol was reviewed and approved by the Institutional Animal Care and Use Committee (IACUC) of Chi Mei Medical Center, Tainan, Taiwan [105122621], on 26 December 2016 by J.J. Wang (IACUC Chairman). In this study, five Duroc Chinese native piglets with an average weight of 25 kg were used for the in vivo studies. Atropine 0.05 mg/kg and tiletamine-zolazepam 6 mg/kg were given intramuscularly for the induction of general anesthesia. These animals were intubated, ventilated, and then maintained with isoflurane (inhalation anesthetic). The anterior axillary line area of these piglets was dissected. We separated every visible layer from the skin, fat, muscle, subclavian artery, subclavian vein, lung, and pleural cavity, as shown in Figure 3. The optical probe (including two laser diodes and one photodiode) was held by the operator. Then, the received voltage signal was digitized and sent to the wireless transmission circuit to transmit to the host system wirelessly via Bluetooth. The program in the laptop analyzed these data in real-time. We simulated the route of the introducer needle layer by layer when we performed the procedure of SCV catheter cannulation. Here, a 14-gauge introducer needle (Arrow, 5.5 Fr, Blue FlexTip® Catheter) embedded with optical fibers was used. We collected absorbed light parameters from the skin to the target vessel in every tissue layer for about ten seconds. An analysis of variance (ANOVA) was used to analyze experimental trials, and differences with p-values <0.05 were considered statistically significant.

## 3. Results

### 3.1. Hemoglobin Parameter of Different Tissue Components

In this section, we investigate the changes in optical density of different tissue components corresponding to different wavelengths. Figure 4a,b show the average changes in optical density of different tissue components at 690 nm and 850 nm light, respectively. At a 690 nm wavelength, the change in optical density of venous blood was greater than that of arterial blood. However, at an 850 nm wavelength, the change in optical density of arterial blood was greater than that of venous blood. Moreover, at both 690 nm and 850 nm wavelengths, the changes in optical density of the muscle tissue were greater than those of the fat, skin, and lung tissue. The changes in the optical density of lung tissue at 690 nm and 850 nm were the smallest, and those of pleural cavities at these wavelengths were the greatest. Moreover, the differences between the changes in optical density of different tissue components at 690 nm (p = 0.000) and 850 nm (p = 0.000) were significant.

Figure 5a–c show the average $HbO_{2}$ and Hb concentrations, and the $StO_{2}$ of different tissue components, respectively. The relative $HbO_{2}$ concentration and $StO_{2}$ of venous blood was lower than that of arterial blood, but the relative Hb concentration of venous blood was higher than that of arterial blood. All of the relative $HbO_{2}$ and Hb concentrations and $StO_{2}$ of arterial blood and venous blood were higher than that of muscle, fat, skin, and lung tissue. Compared with fat and skin tissues, muscle tissue contained higher hemoglobin concentrations. Moreover, the relative hemoglobin concentration of lung tissue was the lowest, and that of the pleural cavity was the highest. Furthermore, the $StO_{2}$ values of arterial and venous blood were higher than that of other tissues. Additionally, the differences between the relative $HbO_{2}$ (p = 0.000) and Hb (p = 0.000) concentrations, and the $StO_{2}$ (p = 0.000) of different tissue components, were significant.

### 3.2. Performance of RBFNN in Recognizing Human Tissue Components

In this study, RBFNN was used to recognize types of human tissue. In the training stage, the target signal of the RBFNN output for the blood vessel group (artery and vein) was set to 1, and it was set to 0 for the other tissue layers (skin, fat, muscle, lung, and pleural cavity). If the RBFNN output was higher than the threshold, it manifested as the blood vessel group. If the RBFNN output was lower than the threshold, it manifested as the other tissue group. Consequently, the RBFNN output could also be read as an index associated with the type of human tissue components. Here, based on optical density changes at 690 nm and 850 nm, we used $HbO_{2}$ and Hb concentrations as the input of RBFNN. We then determined the optimal threshold of RBFNN. To assess the classification performance, we had to define several parameters for the binary classification test: true-positive (TP: the blood vessel group is properly recognized as the blood vessel group), false-positive (FP: the other tissue group is improperly recognized as the blood vessel group), true-negative (TN: the other tissue group is properly recognized as the other tissue group), and false-negative (FN: the blood vessel group is improperly recognized as the other tissue group). In this study, we used F-measure, which is the harmonic mean of the positive predictive value (PPV), and sensitivity for assessing the classification performance, and it can be represented by:(7)$$f-measure=2\cdot \frac{PPV\cdot sensitivity}{PPV+sensitivity}$$

The threshold of RBFNN was set from 0.1 to 0.9 in the training stage. A total of 300 trials were used for training. In the training stage, the optimal F-measure value was 92.86% (precision = 100% and recall = 86.67%) when the neuron number of the hidden layer and the threshold were set to 64 and 0.5, respectively. Next, a blind test (hidden neuron number = 64, threshold = 0.5) was performed, and a total of 100 trials were used for a blind test. In the blind test stage, the F-measure value and accuracy were 93.02% and 94%, respectively (precision = 86.96%, recall = 100%).

### 3.3. Comparison to the Commercial Products

The comparisons between the proposed system and other systems, which may be applied in assisting cannulation, are summarized in Table 1. The vascular transillumination device (Vein Viewer) [27] is a system that can provide real-time two-dimensional projection of superficial vessels onto the skin. Using this system, a 740 nm light source is applied, and a charged couple device camera is used to receive the light reflected from the skin. However, the limitation of this system is that it cannot monitor the deeper vessels. Clinically, the ultrasound system (HD11 XE) [28] is one of the most commonly used systems for assisting cannulation. It can effectively display the structure of muscle and soft tissues, but it is relatively difficult to image bone tissue and gas-rich tissues. It also contains several limitations in the viewing angle of the tissue layer, in particular, when performing SCV cannulation. This is due to ultrasound having difficulty in penetrating the clavicle and ribs. The optical coherence tomography system (SD-OCT 5000) [29] can be used to image the tissue structure surrounding the OCT probe, and it has the advantage of higher image resolution. However, this technique can only assist the physician in distinguishing the tissue structure, and the limitation is the depth. OCT is a capable system which can probably give reasonable results for our application. However, it is a more complicated and expensive system than what is presented in this paper.

## 4. Discussion

Figure 4a,b show that the change in optical density of the veins at 690 nm was greater than that of the arteries. However, the change in optical density of the veins at 850 nm was smaller than that of the arteries. The above results support the phenomenon that the absorption spectra of oxy-hemoglobin and deoxy-hemoglobin coincide at about 800 nm. The change in optical density of muscle tissue was greater than that of fat tissue, and that of the fat tissue was greater than that of skin tissue. This phenomenon fits the experimental results in previous studies, i.e., that the absorption coefficient of fat tissue is greater than that of the skin tissue [30], and that of muscle tissue is greater than that of fat and skin tissues [31]. The change in optical density of the lung tissue was the smallest. Figure 5a–c show that the $HbO_{2}$ value of the artery was higher than that of the vein, the Hb value of the vein was higher than that of the artery, and the $StO_{2}$ value of the artery was higher than that of the vein. This may be attributed to the fact that most arteries contain oxygenated blood and most veins contain deoxygenated blood. The information on the relative hemoglobin concentrations also indirectly reflects the blood vessel distribution densities of the skin, fat, and muscle, owing to the association between relative hemoglobin concentration and tissue blood volume. The relative $HbO_{2}$ and Hb concentrations and the $StO_{2}$ of the lung tissue were also lower than those of other tissues. Lung tissue is full of pulmonary alveoli [32]. When light penetrates through human tissue, the penetration rate of lung tissue may be higher than that of other tissues due to its structure being relatively less dense, which may also result in a lower hemoglobin concentration. Moreover, this is due to the lung tissue showing the lowest optical density change and having the lowest hemoglobin concentration. The pleural cavity presents the maximum optical density change at both 690 nm and 850 nm. As the pleural cavity is a cavity [33], it is expected that most of the light will directly penetrate through thin fluid-filled space, with only a few photons being absorbed, scattered, or reflected by the tissues of the pleural cavity. Therefore, the highest $HbO_{2}$ and Hb concentrations measured in the tissue of the pleural cavity may be inaccurate. The highest estimated $HbO_{2}$ and Hb concentrations in the tissue of the pleural cavity may be due to only a few scattered or reflected photons being received by the detector of the optical probe. According to the significant difference between the absorption spectra and blood vessel distribution of different types of human tissue, the neural network technique was also applied in the classification of the artery, vein, and other types of tissue. From the experimental results, the proposed smart assistance system performed excellently (accuracy = 93.02%) in terms of recognizing blood vessels and other tissues.

We positioned the ultrasound probe in a perpendicular direction to the vessel (a short-axis view), as shown in Figure 6a. It offered the operator a good midline anatomy and permitted an “out-of-plane” needle-guided approach to the target vessel. However, the ultrasound beam crossed the needle shaft and it was difficult to find the needle tip. In Figure 6b, the needle tip is invisible. SCV cannulation without visualizing the needle tip is dangerous. Alternatively, a long-axis view (Figure 6c) was obtained with the probe and vessel axes in the parallel direction. Aiming between a one-millimeter thickness of the ultrasound beam and one-millimeter thickness of the needle is challenging, where the needle shaft and tip might be mismatched in front of or behind the narrow ultrasound beam in the deep three-dimensional space. Compared with the above methods in Table 1, the proposed system could effectively recognize deeper arteries, veins, and other tissues surrounding the optical probe, but the spatial resolution of the proposed system was also poor.

There were some limitations in this study. First, photons might be absorbed by hematomas caused by trauma to the main branch of the vessels. We carefully avoided injury to the vessels; however, sometimes, we could not completely prevent this. Second, the lag-time of the analysis on the laptop was about two seconds. The RBFNN was a useful tool for recognizing the various tissue layers; however, the algorithm took some time to complete in the laptop. This problem may be resolved by an upgraded laptop and a simplified algorithm in the future.

## 5. Conclusions

In this study, a novel smart assistance system for blood vessel catheter placement was proposed. By using the difference of the absorption characteristics and relative hemoglobin concentrations in different tissues, the type of tissue near the needle tip could be estimated indirectly. According to the animal experimental results, the type of human tissue (artery, vein, muscle, fat, skin, lung, and pleural cavity) significantly reflected its absorption characteristics and the change in relative hemoglobin concentrations. Blood vessels and other human tissue groups could also be effectively distinguished by the proposed smart assistance system. Compared with other methods, such as the vascular transillumination device, ultrasound system, and optical coherence tomography system, the spatial resolution of the proposed system was poorer, but it could effectively distinguish between deeper tissues. Therefore, the proposed system has great potential for guiding the physician during catheter placement.

## Figures and Tables

**Figure 1 sensors-20-01891-f001:**
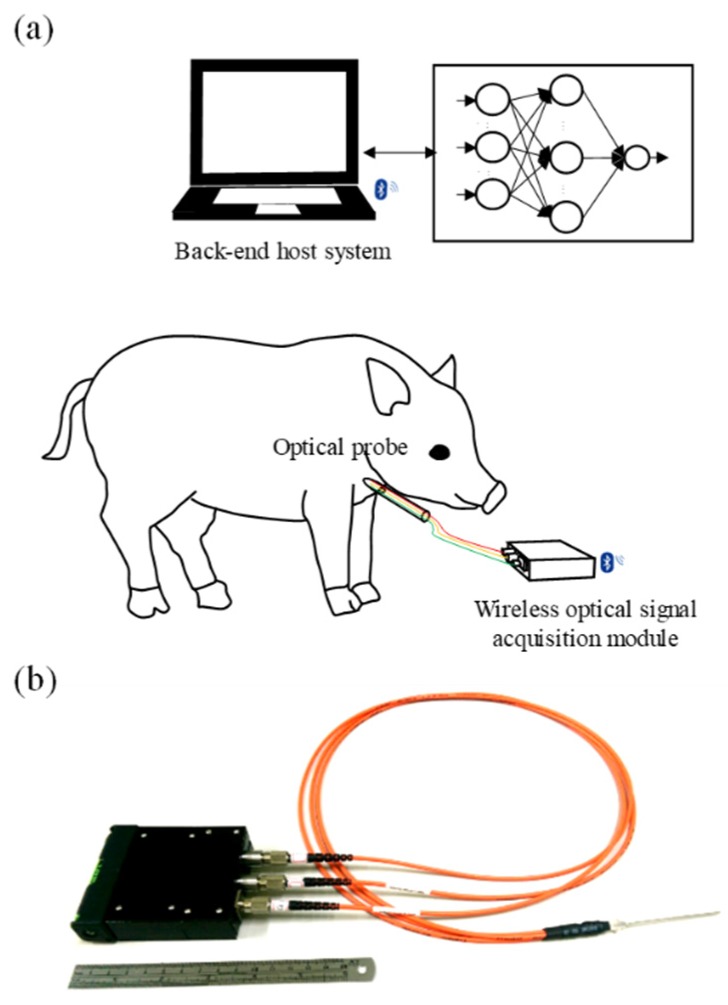
(**a**) Basic schematic and (**b**) photograph of proposed assistance system for blood vessel catheter placement.

**Figure 2 sensors-20-01891-f002:**
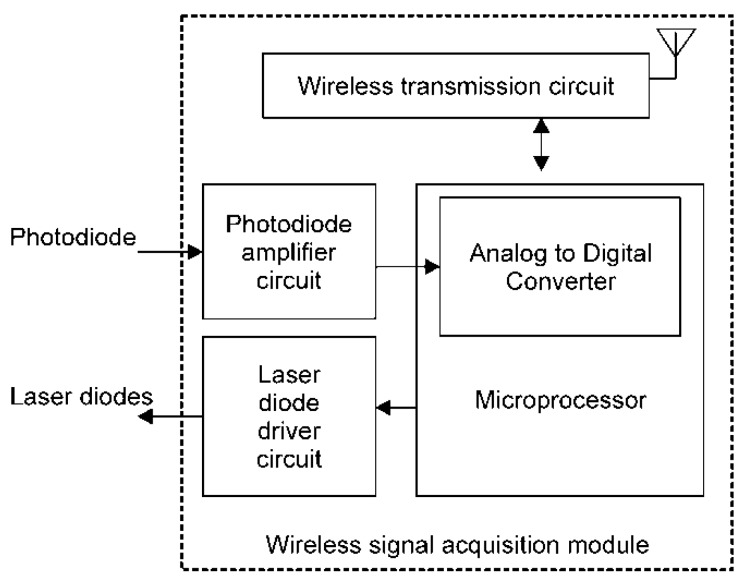
Block diagram of the wireless optical signal acquisition module.

**Figure 3 sensors-20-01891-f003:**
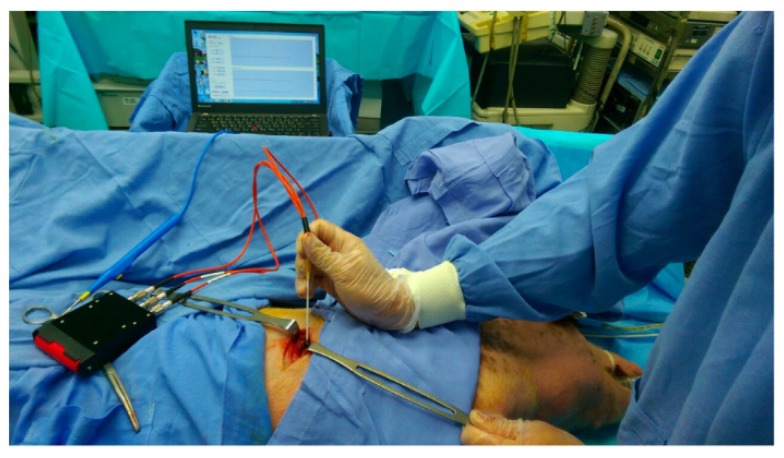
Photograph of the proposed catheter placement assistance system being used during the animal study. The piglet anterior axillary line area was dissected layer by layer from the skin, fat, muscle, subclavian artery, subclavian vein, lung, and pleural cavity.

**Figure 4 sensors-20-01891-f004:**
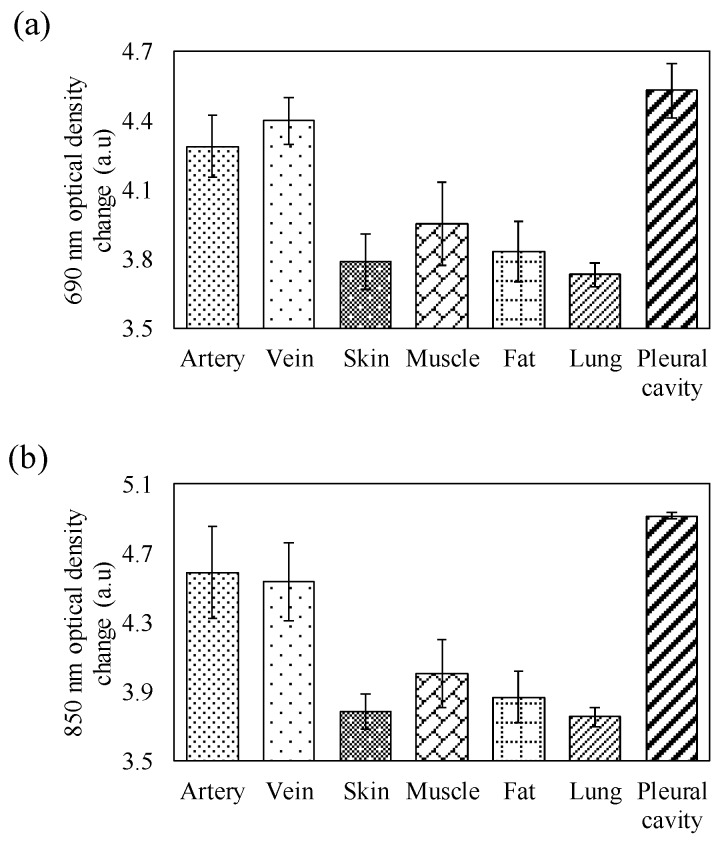
Average optical density changes of different tissues corresponding with (**a**) 690 nm and (**b**) 850 nm wavelengths in the in vivo experiment.

**Figure 5 sensors-20-01891-f005:**
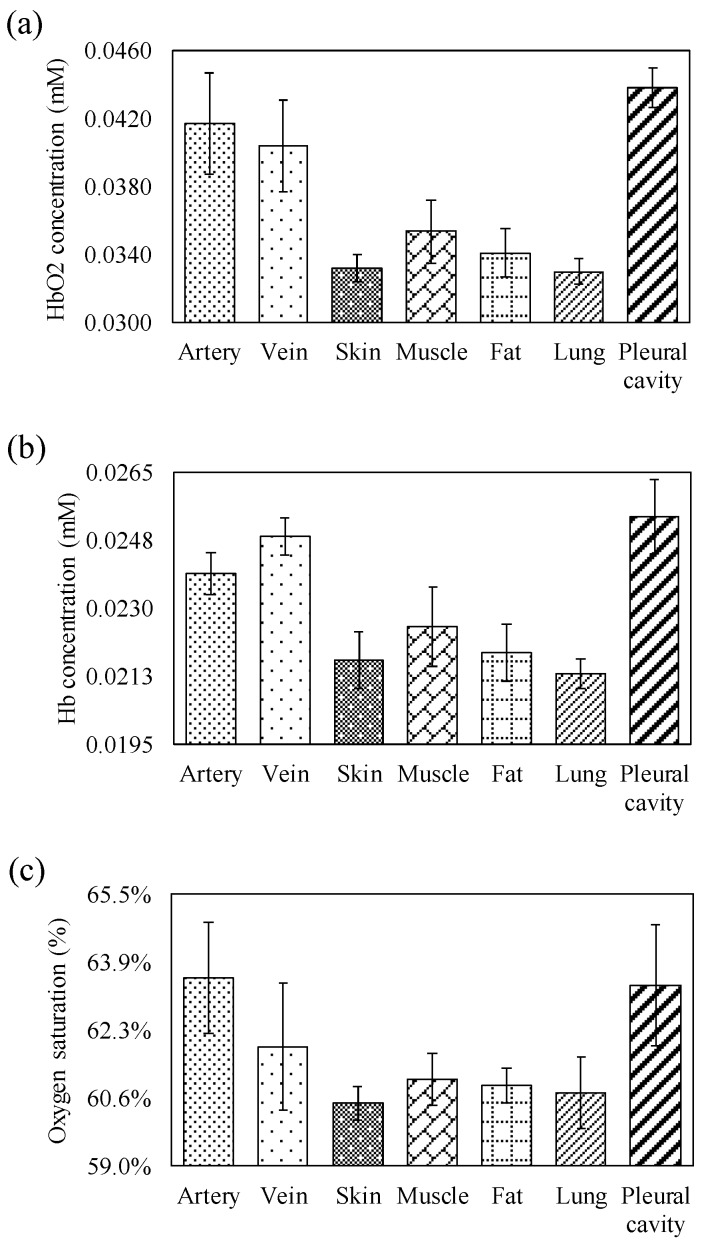
(**a**) Average $HbO_{2}$ concentrations, (**b**) Hb concentrations, and (**c**) $StO_{2}$ of different tissues in the in vivo experiment.

**Figure 6 sensors-20-01891-f006:**
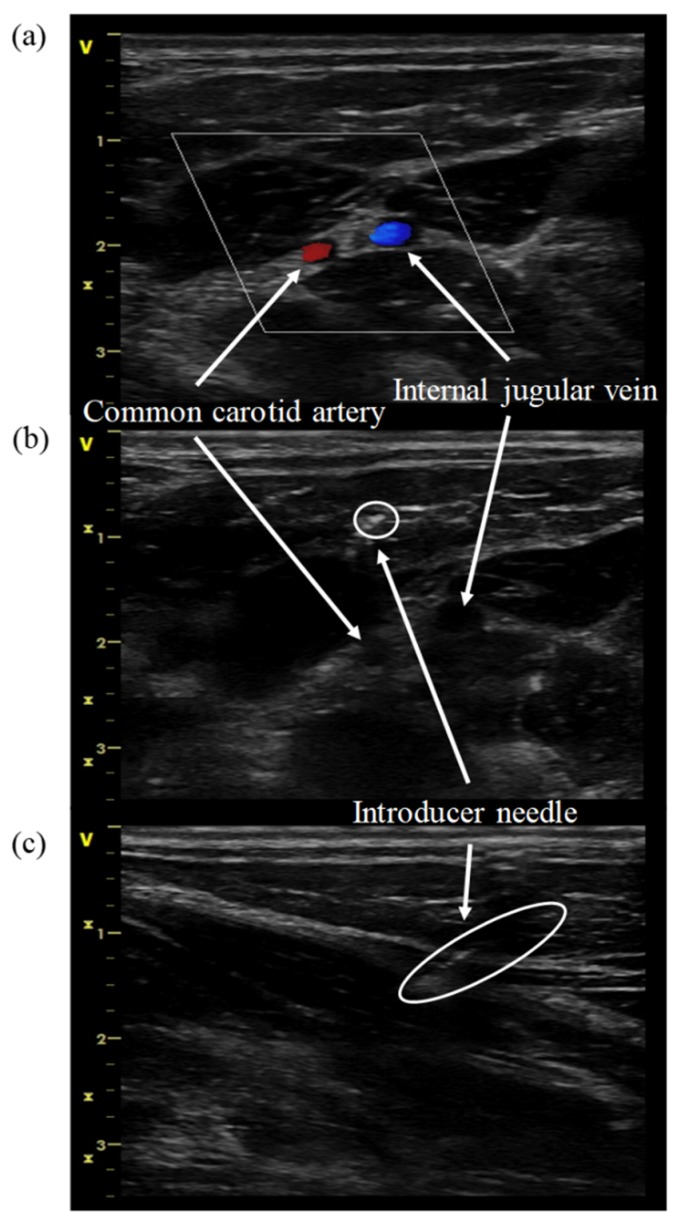
(**a**) Short-axis view of the internal jugular vein; (**b**) out-of-plane approach in the short-axis view of the internal jugular vein, with a mismatch between the ultrasound beam and introducer needle tip; and (**c**) in-plane approach, in long-axis view, of the internal jugular vein. The ultrasound beam is not completely parallel to the needle shaft, and only part of the needle shaft can be seen.

**Table 1 sensors-20-01891-t001:** System comparison between the proposed system and other systems.

	Vein Viewer [27]	HD11 XE [28]	SD-OCT 5000 [29]	Proposed System
Sensing technique	Near-infrared spectroscopy	Ultrasound	Optical coherence tomography	Near-infrared spectroscopy
Sensor type	CCD camera	Ultrasound Probe	OCT probe	Optical probe
Channels	1	1	1	1
Transmission mode	-	USB	USB	Bluetooth
System size (cm^3^)	4.8 × 6 × 19.8	53 × 110 × 151	65 × 46 × 53	11 × 7.5 × 2.5
Wavelength (nm)	740	-	840	690, 850
Physiological parameters	2-D image	2-D image	3-D image	$StO_{2}$, hemoglobin concentration
System complexity	Low	High	High	Low
Advantages	Distinguishability of vessel types	Imaging capability of soft tissue structure	Imaging capability of tissue structure; higher image resolution	Distinguishability of vessel types in deeper tissue
Limitations	Depth limitation	Bone; air; needle tip recognition	Depth limitation	Low image resolution

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
