# Peer review of "A Novel Smart Assistance System for Blood Vessel Approaching: A Technical Report Based on Oximetry"

_sensors, 2020, doi:10.3390/s20071891_

Round 1

Reviewer 1 Report

In this paper, the authors present a novel smart assistance system for blood vessel catheter placement. They tried to explain this system from the theoretical and experimental methods. The theory, experimental setup and results of this paper are complete. The idea of the work is interesting and attractive. This paper has some minor problems and I recommend the following point need to be clarified in the manuscript.

  1. As a complete smart assistance system, does the value (Hemoglobin parameter) measured by the device mentioned in this paper need to be compared with the measured value of commercial products on the market to verify the reliability of the results?

Author Response

Response to Reviewer 1 Comments

Point 1: As a complete smart assistance system, does the value (Hemoglobin parameter) measured by the device mentioned in this paper need to be compared with the measured value of commercial products on the market to verify the reliability of the results? 

Response 1: We thank the reviewer for the comment. We made a comparison table in the discussion of our manuscript. Four different techniques are compared in table 1. In our study, we did not compare our proposed system to commercial products on the market to verify the reliability of the results. It is a good idea to compare the commercial products in another manuscript to validate the reliability and accuracy of the four different techniques.

Reviewer 2 Report

The manuscript suggests a fiber optical based method for catheter placement. A compact system is used for measurements. The optical density were used to differentiate tissue types by measuring hemoglobin content. The study considered measurements on an anaesthesized piglet. The paper is fairly well written and easy to follow. My comments are as below:

  • The authors have performed the measurements invasively. Is the aimed at clinical application also an invasive procedure? If so, could you clarify this in the introduction. Also consider to describe the clinical procedure more clearly in the introduction. 
  • Line 23, you don't need to mention ANOVA in the abstract.
  • In the abstract, line 25, the authors mention in vitro experiments. Are any such results presented? 
  • Please add the wavelength specifications of the light sources on page 3, lines 76-77
  • What is the role of the photodiode? Please be clear on the light sources and wavelengths. 
  • Please be clear on the detection unit and the detected signals. If you are detecting a spectrum, that could be presented in a fig. 
  • Page 4, please mention which wavelengths are used for the analysis
  • Page 4, how many measurements were performed? 
  • Line 157, what do you mean by 'embedded'?
  • Table 1 should be moved to earlier than discussion
  • Is there any point with including Fig. 3 as it is not supportive of any methods in this paper? It should however not be in the discussion. 
  • Lines 273-274, what do you aim to mention by this sentence? OCT is a capable system which probably can give reasonable results for your application. However, it is a more complicated and expensive system than what is presented in this paper.

Author Response

Response to Reviewer 2 Comments

Point 1: The authors have performed the measurements invasively. Is the aimed at clinical application also an invasive procedure? If so, could you clarify this in the introduction. Also consider to describe the clinical procedure more clearly in the introduction.

Response 1: We thank the reviewer for the comment. The original manuscript has been revised as your comment. In clinical application, this is an invasive procedure, whereby we first identify the central blood vessel (most commonly the internal jugular, subclavian, or femoral vein) and then  insert a catheter into it.

Point 2: Line 23, you don't need to mention ANOVA in the abstract.

Response 2: We thank the reviewer for the comment. The original manuscript has been revised as your comment.

Point 3: In the abstract, line 25, the authors mention in vitro experiments. Are any such results presented?

Response 3: We thank the reviewer for the comment. The results of the in vitro experiment were too messy to present. We deleted the words in line 25 in vitro. The original manuscript has been revised as your comment.

Point 4: Please add the wavelength specifications of the light sources on page 3, lines 76-77

Response 4: We thank the reviewer for the comment. The optical probe consists of two laser diodes with wavelength 690 nm or 850 nm. The original manuscript has been revised as your comment.

Point 5: What is the role of the photodiode? Please be clear on the light sources and wavelengths.

Response 5: We thank the reviewer for the comment. The design of the photodiode amplifier circuit is based on a trans-impedance amplifier; it can convert the photocurrent generated from the photodiode to a voltage and amplify this voltage. The photodiode is the detector of the optical probe, which received a few scattered or reflected photons from two laser diodes. We add these statement in our manuscript. 

Point 6: Please be clear on the detection unit and the detected signals. If you are detecting a spectrum, that could be presented in a fig.

Response 6: Many thanks for the reviewer’s comment. The light signal acquired by the photodiode was a current signal, and the current signal was amplified and converted into a voltage signal. Therefore, the unit of the light signals  and  was voltage. In this study, the dual wavelength light source was cross-switched by the microprocessor. Therefore, we received one wavelength light signal at a time.

Point 7: Page 4, please mention which wavelengths are used for the analysis

Response 7: We thank the reviewer for the comment. When photons with wavelength 690 nm or 850 nm penetrate the tissue layer, some of them will be absorbed or scattered to cause light attenuation. The photodiode received a few scattered or reflected photons. These light signals were cross-switched by the microprocessor and used for analysis. The original manuscript has been revised per your comment.

Point 8: Page 4, how many measurements were performed?

Response 8: We thank the reviewer for the comment. In this study, five Duroc Chinese native piglets with an average weight of 25 kg were used for the in vivo studies. We collected absorbed light parameters from the skin to the target vessel in every tissue layer for about ten seconds.

Point 9: Line 157, what do you mean by 'embedded'?

Response 9: We thank the reviewer for the comment. We embedded these optical fibers (with two laser diodes and one photodiode) into a 14-gauge introducer needle (figure 1 b). These optical fibers are embedded into the needle tube, and they can reach the deeper tissue easily via insertion of the needle tube.

Point 10: Table 1 should be moved to earlier than discussion

Response 10: Many thanks for the reviewer’s comment. We moved table 1 to the end of the result’s section.

Point 11: Is there any point with including Fig. 3 as it is not supportive of any methods in this paper? It should however not be in the discussion.

Response 11: We thank the reviewer for the comment. In 2.3 Experimental Design, we described the animal used in our study. This picture demonstrated how we used our proposed system in the animal study. The optical probe (including two laser diodes and one photodiode) was held by the operator. Then, the received voltage signal was digitised and sent to the wireless transmission circuit to transmit to the host system wirelessly via Bluetooth. The program in the laptop analyzed these data in real-time. The original manuscript has been revised per your comment.

Point 12: Lines 273-274, what do you aim to mention by this sentence? OCT is a capable system which probably can give reasonable results for your application. However, it is a more complicated and expensive system than what is presented in this paper.

Response 12: We thank the reviewer for the comment. The original manuscript has been revised as your comment. However, this technique can only assist the physician in distinguishing the tissue structure, and the limitation is the depth. OCT is a capable system which can probably give reasonable results for our application. However, it is a more complicated and expensive system than what is presented in this paper.
